# Spinal Cord Morphology in Degenerative Cervical Myelopathy Patients; Assessing Key Morphological Characteristics Using Machine Vision Tools

**DOI:** 10.3390/jcm10040892

**Published:** 2021-02-23

**Authors:** Kalum Ost, W. Bradley Jacobs, Nathan Evaniew, Julien Cohen-Adad, David Anderson, David W. Cadotte

**Affiliations:** 1Hotchkiss Brain Institute, Cumming School of Medicine, University of Calgary, Calgary, AB T2N 1N4, Canada; kalum.ost@ucalgary.ca; 2Department of Clinical Neurosciences, Section of Neurosurgery, Cumming School of Medicine, University of Calgary, Calgary, AB T2N 1N4, Canada; wbjacobs@ucalgary.ca; 3Combined Orthopedic and Neurosurgery Spine Program, University of Calgary, Calgary, AB T2N 1N4, Canada; Nathan.Evaniew@albertahealthservices.ca; 4Section of Orthopaedic Surgery, Department of Surgery, University of Calgary, Calgary, AB T2N 1N4, Canada; 5NeuroPoly Lab, Institute of Biomedical Engineering, Polytechnique Montrèal, Montrèal, QC H3T 1J4, Canada; jcohen@polymtl.ca; 6Functional Neuroimaging Unit, CRIUGM, Universitè de Montrèal, Montrèal, QC H3T 1J4, Canada; 7Mila-Quebec AI Institute, Montrèal, QC T2N 1N4, Canada; 8Department of Biochemistry and Molecular Biology, Cumming School of Medicine, University of Calgary, Calgary, AB T2N 1N4, Canada; david.anderson1@ucalgary.ca

**Keywords:** degenerative cervical myelopathy, personalized medicine, machine learning, spinal cord

## Abstract

Despite Degenerative Cervical Myelopathy (DCM) being the most common form of spinal cord injury, effective methods to evaluate patients for its presence and severity are only starting to appear. Evaluation of patient images, while fast, is often unreliable; the pathology of DCM is complex, and clinicians often have difficulty predicting patient prognosis. Automated tools, such as the Spinal Cord Toolbox (SCT), show promise, but remain in the early stages of development. To evaluate the current state of an SCT automated process, we applied it to MR imaging records from 328 DCM patients, using the modified Japanese Orthopedic Associate scale as a measure of DCM severity. We found that the metrics extracted from these automated methods are insufficient to reliably predict disease severity. Such automated processes showed potential, however, by highlighting trends and barriers which future analyses could, with time, overcome. This, paired with findings from other studies with similar processes, suggests that additional non-imaging metrics could be added to achieve diagnostically relevant predictions. Although modeling techniques such as these are still in their infancy, future models of DCM severity could greatly improve automated clinical diagnosis, communications with patients, and patient outcomes.

## 1. Introduction

Degenerative Cervical Myelopathy (DCM) is the most common form of spinal cord injury worldwide [1], and is associated with substantial impairment of patient quality of life. DCM manifests in patients as progressively worsening pain, numbness, dexterity loss, gait imbalance, and sphincter dysfunction [2], the result of degenerative compression of the cervical spinal cord. Timely diagnosis of DCM is critically important to minimize neurological deterioration, but is challenging because the symptomatology of DCM overlaps with many other common diseases [3]. DCM symptoms often do not appear until neurological damage has already occurred [4,5], and patients who receive treatment after a longer prodrome of neurological deficits may have worse long-term prognosis [6]. Surgical decompression is the mainstay of treatment, with 1.6 per 100,000 people requiring surgery to treat DCM in their lifetime [7]. In addition to a thorough history and physical examination, routine MRI of the cervical spine is an essential diagnostic test that confirms the presence and extent of spinal cord compression [8].

Once DCM has been diagnosed, patients and their care provides must decide whether to proceed with surgical treatment via surgical decompression. Predictive outcome modeling through computationally aided MRI analysis in this scenario is an attractive possibility, but is currently in its infancy. Current analysis tools include the Functional Magnetic Resonance Imaging of the Brain (FMRIB) Software Library [9], Statistical Parametric Maps [10], and the Medical Image NetCDF format [11]. These tools, however, tend to be generalized and lack the specificity required for spinal cord analyses. Although logistic regression models have been tested and have demonstrated limited success [12], there remains room for improvement. Spinal cord segmentation analysis using qMRI imaging data of patients by tools such as the Spinal Cord Toolbox (SCT) [13] has recently been shown to provide improved predictive power [14], but these tools tend to break down when analyzing damaged spinal cords [15]. Studies which did find success in predicting myelopathic outcomes opted instead to manually inspect the spinal cord [4,16] or manually correct the output of automated analyses [17], reducing the benefits these automated processes provide. To optimize their use, it is imperative to evaluate the extent and source of these limitations. To this end, we assessed the SCT software package for its analytical capabilities in predicting disease severity of DCM. We applied this software package to routinely acquired MRI images from a subset of patients who went on to receive clinical diagnoses of DCM across Alberta, Canada.

## 2. Methods

### 2.1. Computational Tools Used

The program versions for the methods used below were as follows: Spinal Cord Toolbox, v.5.0.1 [13], 3D Slicer v.4.10.2 [18], SciKit-Learn v.0.23.2 [19], SciPy v.1.5.2 [20], matplotlib v.3.3.2 [21], seaborn v.0.11.1 [22], numpy v.1.19.2 [23], and pandas v.1.2.0 [24]. As *CovBat* was still in development at time of this paper’s publication [25], its state at the time of this analysis can be replicated by using the GitHub commit 23a0429, available at https://github.com/andy1764/CovBat_Harmonization/commit/23a0429c2a81e7682da94ff2d0f5e634ab91b429 (accessed on 9 June 2020).

### 2.2. Data Preparation

We identified cervical spine MRI images that were used to diagnose 328 patients with DCM who were serially enrolled in the Canadian Spine Outcomes and Research (CSORN) longitudinal registry (initiated in 2016, ongoing [8]). Data were obtained from multiple clinics across the province of Alberta (Figure 1); each clinic had their own procedures and protocols, resulting in variation in image quality and resolution. This was accounted for, to some extent, via batch effect compensation (see Section 2.4).

Our sample set consisted of a diverse number of imaging methodologies. For example, 257 of our 328 patients records used a magnetic field strength of 1.5T, while the remaining 71 used a field strength of 3T. In general, images were also acquired at a relatively low resolution, with T2 weighted, sagittally oriented images primarily with a center-to-center slice thickness of 3 mm (318 images), 2 mm (52 images), with the remaining images (21 images) ranging from 0.9 mm to 5 mm. Axially oriented T2 weighted images were more diverse, but also relatively low resolution: they primarily consisted of images with a 2.5 mm (164 images), 4 mm (128 images), 3 mm (124 images), and 2 mm (90 images) slice thickness, with the remainder varying between 1.4 mm and 5 mm (54 images).

Digital Imaging and Communications in Medicine (DICOM) data were evaluated, anonymized, and converted into the NIfTI file format, resulting in 1335 total MRI sequences. Imaging files were then manually inspected to confirm data integrity (presence of required files and lack of substantial imaging motion or aliasing), and converted into a BIDS-compliant format [26]. This resulted in 3 patient records and 151 imaging files being excluded, leaving the dataset at 1184 imaging files across 325 patient records. The majority of files dropped were excluded due to excessive noise being present in the image or motion artifacts/patient movement between samples. Other reasons for image exclusion were mislabeling (the MRI images being of the tubular spine, rather than the cervical spine) and insufficient slice count (resulting in the inability for segmentation algorithms to make accurate estimates of spinal cord metrics). Axial images were particularly low quality, making up two thirds of the excluded set (101 of the 151 excluded images).

### 2.3. Spinal Cord Segmentation

Spinal cord segmentation (masking the contents of the spinal cord vs. the other contents of the image) was done manually for a subset of 50 patients, containing a total of 195 images, as to provide a control against automated segmentation techniques (discussed below). These were done via manual inspection across all images by one person using the 3D Slicer application [18].

Automated segmentation for the full set of spinal cord images was then completed using SCT [13]. SCT was selected over its alternatives for two reasons. First, it is the only all-in-one package we are aware of that is specialized for application on the spinal cord, rather than being generalized to MR imaging in general [9,10]. Second, it is well documented and open source, making it easy to use and apply in clinical practices without major legal difficulties or financial burden. SCT provides two primary ways to initially segment the spinal cord; ‘PropSeg’ [27] and ‘DeepSeg’ [28]. PropSeg functions by initially detecting an initial slice of the spinal cord, then propagating that slice across the remainder of the spinal cord, adjusting as it goes. DeepSeg, in contrast, tries to identify the entire segmentation simultaneously, using either a Convolutional Neural Network (CNN) or Support Vector Machine (SVM) to do so. The model can also take into account only data in a given 2D slice, or the entire 3D image; we chose to test all combinations available. This resulted in 5 different automated segmentation methods being assessed in total. A segmentation method comparison, performed on a sagittal MRI image slice from a patient with severe DCM, is shown in Figure 2.

SCT can fail to produce a segmentation outright; there seems to be no discernible trend as to what causes this. In these cases, the segmentation method was simply skipped for the image, with subjects for which all methods failed being excluded. This resulted in 1 patient record being dropped, leaving 324 patients records containing 1066 total images for further analysis.

### 2.4. Metric Extraction and Standardization

Following segmentation, we used SCT’s ‘sct_process_segmentation’ script to extract metrics from each spinal cord image’s segmentations (both automated and manual). All metrics were taken from the entire spinal cord volume, and included the means and standard deviations of the cross-sectional area of the spinal cord segmentation slices (mm squared), anterior/posterior angle (degrees), right/left angle (degrees), anterior/posterior diameter (mm), right/left diameter (mm), eccentricity (ratio of two prior diameter measurements), orientation (relative angle, image to spine), and solidity (ratio of true and convex-fit cross-sectional area). The total length of the spinal cord (mm) was also obtained, being produced by the same analysis pipeline; given its tenuous-at-best relation to the morphology associated with DCM, this was kept to evaluate SCT’s options in full. That is to say, we did not expect length (sum) to be useful to any model, but included for the sake of being thorough.

Collected metrics from each automated segmentation were grouped by “imaging methodology” (the combination of segmentation method, MRI contrast, and MRI orientation) and joined with their respective patient’s modified Japanese Orthopedic Association (mJOA) score. The mJOA is a clinician-reported instrument that measures the symptoms and disability of patients suffering from DCM, whereby lower mJOA scores indicate greater impairment and worse disease severity. It is the recommended and most commonly used metric to assess disability caused by DCM [29]. Scores can range from 18 (healthy) to 0 (inability to move hands or legs, total loss of urinary sphincter control, and complete loss of hand sensation). mJOA scores are also classified categorically as mild (a score of 15 or greater), moderate (a score of 12 to 14), or severe (a score or 11 or less) [30].

We then opted to harmonize the data to remove any effects unique to each scanner in our sample set. This was done using the *CovBat* harmonization program [25], grouping the data by scanner used to acquire it. The scanner of a given image was determined from the DICOM headers of the images, similar to the methods used in the original assessment of the *CovBat* program [25]. Specifically, images were deemed to share the same scanner if they shared the same scanner manufacturer, scanner model, and magnetic field strength. Please note that geography was *not* accounted for, unlike in Chen et al.’s [25] original presentation of the tool. This was because per clinic differences in how the scanner was operated were assumed to be minimal, given the shared health care zone all data was collected within. Not filtering by geography also has the convenient side-effect of keeping our dataset nearly completely intact, as the *CovBat* harmonization process requires that at least 3 elements exist in every group; only one methodology failed to reach this count, leading to only 2 segmentations total being lost. Thus, all patients and images remaining from prior filters remained represented in at least one methodology in the resulting set.

### 2.5. Model Metric Selection

External non-image derived metrics (such as age, sex, and other demographic information) were available, but were intentionally left out from both the data preparation processes prior and the data modeling below. This was to allow our models to evaluate the predictive merit of current automated image processing techniques, without external bias from said parameters. It has already been established that external metrics such as patient demographics are partially effective at predicting DCM severity in patients [31], and creating a composite model runs the risk of over-fitting the data and reducing diagnostic power.

Prior to fitting each model to their associate methodology dataset, data were grouped by the associated image’s acquisition contrast (T1w, T2w, or PDw), segmentation method (options listed prior), and imaging orientation (axial, sagittal, or coronal); the resulting combination is referred to as the “assessment methodology” from this point forward. Initially, as a result of the combinations of these categories, there were potentially 45 different assessment methodologies, though only 30 of these were actually present in our data set. Assessment methodologies with fewer than 3 samples were dropped from the data set, as their lower sample size could lead to inaccurate or misleading results. This resulted in 3 further assessment methodologies being dropped, leaving 27.

Before fitting to models, each assessment methodology was then processed using False Discovery Rate Feature Selection via SciKit-Learn’s SelectFdr function. The scoring function was set to the F-test score of the metric to the mJOA score (evaluated with SciKit-Learn’s ‘f_regression’ function) or DCM severity category (evaluated with SciKit-Learn’s ‘f_classif’ function). The F-test was selected for its ability to evaluate whether data would conform well in a regression model; as we kept to simple regression-based models for this study (see below), this fit our use case perfectly. The allowable probability of false discovery was set to p=0.05. This feature selection process served both to reduce the list of spinal cord morphological metrics to only those anticipated to be correlated with our target metric (our mJOA score or the mJOA severity categories), but also to filter out assessment methodologies which are likely to be ineffective (by selecting 0 features for them). This resulted in a drastic reduction in valid assessment methodologies, with at most 3 passing this stage per severity category and model type (linear or categorical) and proceeding to the final model assessment.

### 2.6. mJOA Correlation and Categorization Model Assessment

The remaining assessment methodologies were then fit to either SciKit-Learn’s ‘LinearRegression’ model (for linear metric to mJOA score models) or ‘LogisticRegression’ model (for DCM severity classification models). These simple models fit linearly to each parameter, allowing for metrics to be evaluated sans-interaction effects, and does so very quickly. This made them ideal for rapid, diverse, and simple assessments, perfect for evaluating the SCT derived metrics on their own. All groups were split into train-test groups using 5-fold shuffle split grouping, and cross-validated by fitting the modeling method to each group in turn. Each resulting model’s effectiveness was then evaluated using $r^{2}$ for the linear regression models, and using receiver operating characteristic area under curve (ROC AUC) for categorical models. The effectiveness of the model type was then assessed via the mean score of all resulting models. To confirm that the somewhat experimental CovBat method worked correctly, all processes prior were run on both the standardized-only metric sets and the CovBat-harmonized metric sets as well. Categorical imbalance was also evaluated for each model type via assessing the accuracy of a “dummy” model, which simply guessed the most common category at all times.

## 3. Results

### 3.1. Spinal Cord Metrics of DCM Patients by mJOA Severity

Overall, with human-derived segmentation methods, very few metrics demonstrated significant differentiation by mJOA severity class, with only derived mean area, mean diameter (along both orientations), and anterior-posterior variance showing such distinction. A summary table of these metrics can be found in Table 2, with a visualized distribution with statistical annotations presented in Figure 3. This suggests that most metrics are not, on their own, sufficient to distinguish between the various mJOA severity classes, let alone predict the mJOA score accurately.

### 3.2. Manual vs. Automated Segmentation Metrics

All the automated segmentation methods were then compared to the manual method to determine whether significant differences existed via one-way ANOVA. This allows us to assess whether statistically significant differences in data distribution existed between our automation derived and manually derived imaging metrics. If such a difference is found to exist, it suggests that the automated process differs in some meaningful way, which may in turn become useful for predicting DCM score and/or mJOA severity. A summary of these metrics can be found in Table 3, with the distributions of said metrics shown and statistically assessed in Figure 4. In summary, the majority of metrics were found to be functionally distinct when measured automatically compared to manually, with the exceptions being eccentricity (both mean and standard deviation) and solidity (both mean and standard deviation). No automated segmentation method appeared to replicate the measures observed with manual methods for all metrics; these deviations could potentially prove useful, however, if how they differ from the manual segmentation method is diagnostically predictive.

### 3.3. mJOA Score Regression by Assessment Methodology

To assess whether the observed patterns of difference represented diagnostically relevant variation, each metric within each assessment methodology (segmentation algorithm, image contrast, and image orientation) was evaluated for significant regression with patient mJOA score (the distribution of which is shown in Figure 5). Of the metrics extracted from the segmentations, almost every metric was found to be significantly predictive (p≤0.05) of a patient’s mJOA score for at least one assessment methodology (evaluated via SciKit-Learn’s ‘f_regression’ function). However, only the T2w contrast, sagittal orientation, and the svm deepseg segmentation algorithm methodology produce a model which had more than 3 parameters significantly related to mJOA score, with 5 total; mean of spinal cross-sectional area (p=0.007), mean of anterior/posterior cross-sectional diameter (p=0.001), mean right/left spinal angle (p=0.024), mean eccentricity (p=0.031), and mean solidity (p=0.013). For all other groups, a combination of these metrics, with the occasional standard deviation of solidity, angle, or diameter was observed to have significant predictive power with the mJOA score. Notably, however, the T2w contrast, sagittal orientation, propseg segmentation algorithm methodology was the only one to find total summed length of the spinal cord as significantly related, despite our assumption that it would not be found as such. A more detailed overview of the distributions of these *p*-values has been visualized by metric (Figure 6) and methodology element (Figure 7).

### 3.4. Linear mJOA Prediction Models

Despite the results prior, none of the assessment methodology models tested produced a multi-parameter linear model that even came close to being remotely accurate, with all performing worse than a ‘dummy’ random chance-based model ($r^{2}=0$). The $r^{2}$ scores for each were evaluated by SciKit-Learn’s ‘r2_score’ function, which can produce negative $r^{2}$ scores which imply that the associated model is worse-than-random. For non-batch compensated data, the $r^{2}$ scores hovered around −30, while batch compensated metric derived models resulted in $r^{2}$ scores ranging from −25 to −10. False Discovery Rate Feature Selection also tended to choose more features for the harmonized data set (with harmonized models having an average of 2 features selected, versus the 1.33 feature average form models trained on standardized metrics alone). This implies that the harmonization processed removed noise which otherwise masked useful trends, though clearly this was still not enough to lead to a valuable model. Tables summarizing these attributes, for both standardized (Table 4) and harmonized (Table 5), are available for further inspection.

### 3.5. Logistic DCM Categorical Models

Overall, the categorization models proved far more effective, with one reaching an ROC AUC of 0.92 (sagittal PDw 3d SVM deepseg methodology, not harmonized), with an average ROC AUC of 0.654 for non-harmonized data trained models and 0.612 for CovBat-harmonized data trained models. The mild mJOA model proved best overall, followed by the severe mJOA model and, finally, the moderate mJOA model. Models with fewer samples also tended to have higher ROC AUC scores, suggesting some level of over-fitting was occurring, as the higher sample count provided more natural noise which the models could erroneously detect as significant. The full results are summarized in Table 6 (non-harmonized) and Table 7 (CovBat-harmonized).

## 4. Discussion and Conclusions

In this work, we explored predictive outcome modeling using computationally aided MRI analysis. We attempted to extract metrics used by trained surgeons from MRI images of the human cervical spine to predict disease severity. Most of these derived metrics simply lack sufficient differentiation across mJOA score severity. Variation appears to be mostly patient-specific rather than related to DCM severity. This is likely a result of the metrics being sampled across the entirety of the spinal cord, whereas morphological differences related to DCM often only effect a portion of the spinal cord, with the remainder appearing ‘healthy’. Although there were some interesting trends within the data, these useful trends appear to be masked by natural inter-individual variance between each of the patients enrolled in this study. As a result, our machine learning systems had difficulty pulling out said meaningful trends, resulting in over-fitting to patient variation and lower overall accuracy.

Non-imaging metrics, such as age, smoking status, and symptom duration have been shown to be important metrics in the development of models to predict patient outcomes after surgical treatment for DCM [32]. MR imaging of the cervical spine plays a vital role in the diagnosis and surgical treatment planning of this patient population. Although this data is vital to a surgeon’s decision-making process, most surgeons would not consider treating a patient without and MRI confirmed diagnosis. Efforts to distill a surgeon’s acumen into an ‘imaging metric’ have fallen short in terms of predictive capabilities. Our work, while novel in computational approach, only adds to this body of literature, bringing us closer to integrating advanced imaging metrics with a patient’s clinical presentation. Such a reality could greatly improve a surgeon’s ability to treat their patients.

The models we presented in this work highlight some key features which we can use to inform future processes. Given the low accuracy of most assessment methodologies, the vast majority of metrics extracted from these segmentations did not correlate strongly with mJOA scores. However, a handful did, showing that assessment methodologies could identify statistically significant correlations. Spinal cord segmentation metrics chosen via feature selection also showed an interesting trend, with the angle and diameter of the spine being selected most commonly, followed by metrics associated with cross-sectional area and spinal cord solidity/eccentricity. This is unsurprising given that pathology of DCM results in compression of the spinal cord (i.e. reduction in diameter, often resulting in a misshapen cross-section), but it nonetheless highlights the potential for a model which focused solely on identifying key variations in these values derived directly from the image itself. It is plausible that finding a way to normalize these metrics relative to the patient’s unique spinal cord variations could be incredibly valuable for creating a diagnostic model. These techniques show potential, but appear to be hampered by the natural variance of DCM patients’ spinal cords.

There are several limitations to this study. First, all data comes from central-southern Alberta (Figure 1), potentially leading to some implicit demographic attributes of the region influencing the analyses. Second, only relatively simple models (Linear and Logistic regression) were used, whereas more complex models may have proven more useful. Simple models simply cannot capture any significant interaction effects. Given the complexity of DCM, it is extremely likely at least one such severity influencing ‘complex’ effect exists. We limited our analyses to these simpler models to focus the study on evaluating major trends in the data to inform future model design. Third, only simple measures of accuracy were used ($r^{2}$ simply assesses a model’s total explained variance, whereas ROC AUC measures its relative ability to predict true positives over false positives), which are likely to mask important details on how each model functions. More nuanced assessment metrics should be considered for future models aimed at diagnostic application; measurements such as false positive rate vs. false negative rate are likely to be far more significant metrics in these contexts (a false positive will be likely caught and dismissed by a clinician upon review, whereas a false negative could lead to significant health consequences for the patient). Fourth, the cross-validation procedure (5-fold) was chosen for its simple implementation in both linear and logistic regression models. A leave-one-out (linear regression) or leave-one-per-category-out (logistic regression) model would be more appropriate here, as it would replicate how a real-world implementation of similar predictive models would be required to function; with a single new patient record being submitted in varying intervals and predictions made for them. Such cross-validation may result in models more prone to over-fitting noise; however, finding noise-resistant metrics would be a must before this limitation could be resolved. Fifth, we only accounted for metrics directly extracted from MRI images. Prior studies have shown that non-imaging metrics can also influence spinal cord morphometrics within a patient [33], and as a result it is likely some confounding or contributing effect from such non-imaging metrics may have not been accounted for. Finding a way to fold in these metrics could improve future models substantially.

Given these limitations, future studies which aim to model DCM outcomes should aim to identify metrics which are normalized to healthy patient variation. This would reduce the amount new models will overfit to natural patient variation over DCM relevant attributes. Likewise, due in part to the limited number of samples available in our dataset and the fact all were diagnosed with DCM, asymptomatic persons who display traits analogous to those of DCM were not accounted for. Prior work has shown MRI images from asymptomatic persons can appear similar to those taken from DCM patients [34]. Increasing the number of MRIs taken from healthy individuals could reduce the likelihood of future models becoming too liberal with their DCM diagnoses. Finding metrics resilient to these forms of over-fitting is imperative if any resulting model is to be implemented in a fully autonomous manner, as to avoid incorrect diagnostic conclusions which may lead to patient harm.

Several possible solutions exist to address these limitations. First, normalizing metrics to be relative per-patient could greatly mitigate natural patient variance effects. These could include ratio metrics (i.e., minimum over maximum ratio), internal outlier detection (i.e., detecting drastic changes in spinal cord shape relative to the rest of the spine), or even dynamically generated metrics such as those produce by Principle Component Analysis. Such metrics would both provide internal normalization for patients, and (in the case of Principle Component Analysis) would be specifically selected based on their relevance to the DCM severity. Second, experimenting with more complex models stands to capture more nuanced details of DCM, such as those of interaction effects between multiple parameters. This would require said metrics to be refined beforehand, however, as such interaction effects would be particularly prone to natural noise masking true relations. Finally, folding in non-imaging derived metrics could address the issue of ‘asymptomatic’ false positives mentioned prior. Given these effects would likely need to be considered alongside spinal cord morphology metrics, this should be done after the selection of said morphological metrics and after a suitable model is chosen which can reflect these interactions. The outcome of such research could be particularly enlightening, helping to explain what distinguishes asymptomatic persons from those suffering from DCM, potentially providing improved treatment options for the latter.

Overall, it appears that modern computational methods have unmet potential in diagnostic prediction of DCM severity. With improvement of these models via the integration of external non-imaging derived metrics, deploying additional complex statistical and machine learning models, and improved morphological metric identification, it may be possible to create a system capable of working at least as effectively as the average clinician. The numerous limitations of this study will also need to be addressed should such a system come to fruition, namely the problem of models over-fitting to natural patient variation and other noise rather than DCM specific morphological characteristics. If these challenges are met, such a system being integrated in a fully automated capacity could potentially revolutionize the treatment of DCM. Such a system could allow clinicians to focus on each patient’s needs more closely, helping them come to more informed treatment decisions and mitigating risks associated with their chosen treatment. This model could also greatly improve our understanding of DCM, potentially identifying targets for new modes of treatment or discovering novel diagnostic metrics.

## Figures and Tables

**Figure 1 jcm-10-00892-f001:**
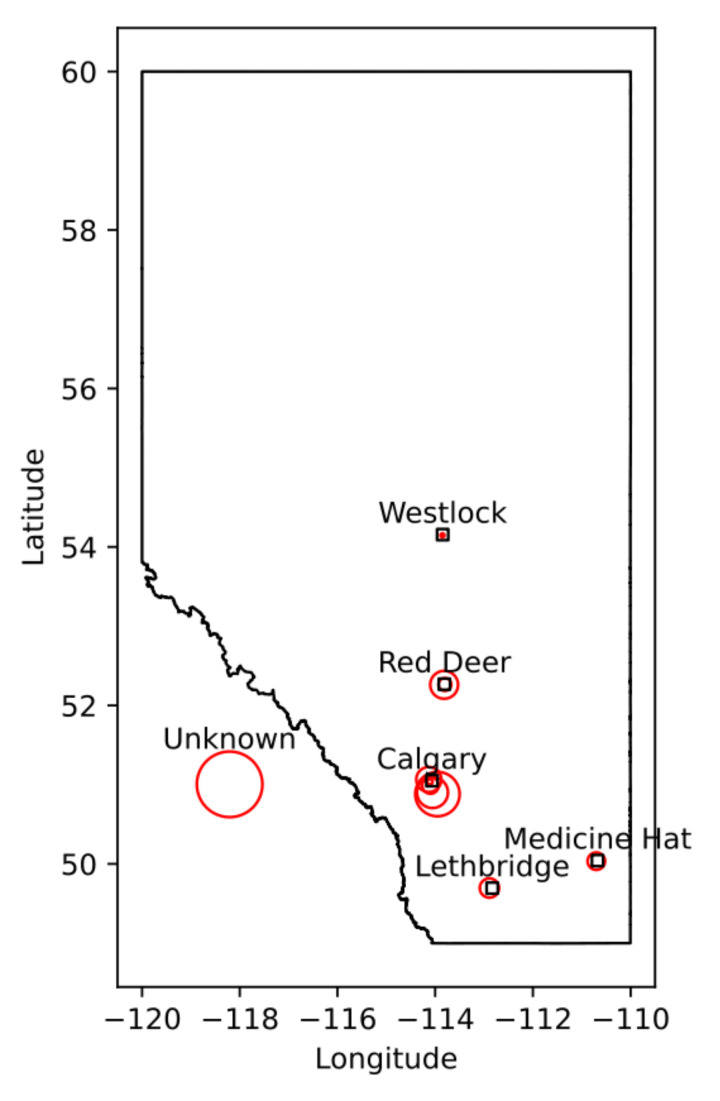
The distribution of clinics in Alberta, as well as their relative contribution of the dataset. Larger circles indicate larger contributions (in number of patients), with each circle representing one clinic.

**Figure 2 jcm-10-00892-f002:**
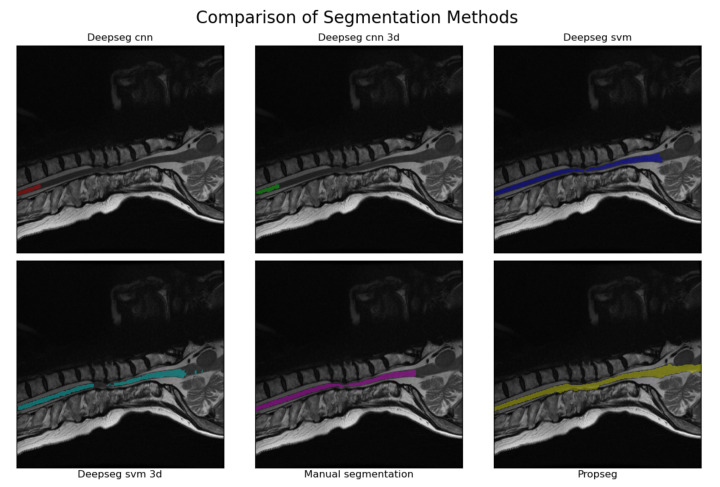
An example of the segmentations produced by each of the methodologies tested. The image used was that of a sagittal, T2w image from a patient with severe DCM (as evaluated by mJOA score). The manually segmented example is provided in the bottom center, with all others being produced via automated analyses using SCT [13]. The CNN kernel in particular seems to struggle when faced with spinal cord compressions, with the SVM kernel and propseg method having relatively minor issues in comparison (usually leaking or outright ignoring the compressed areas instead). This pattern appeared to hold true for all segmentations manually reviewed during the process to create Table 1.

**Figure 3 jcm-10-00892-f003:**
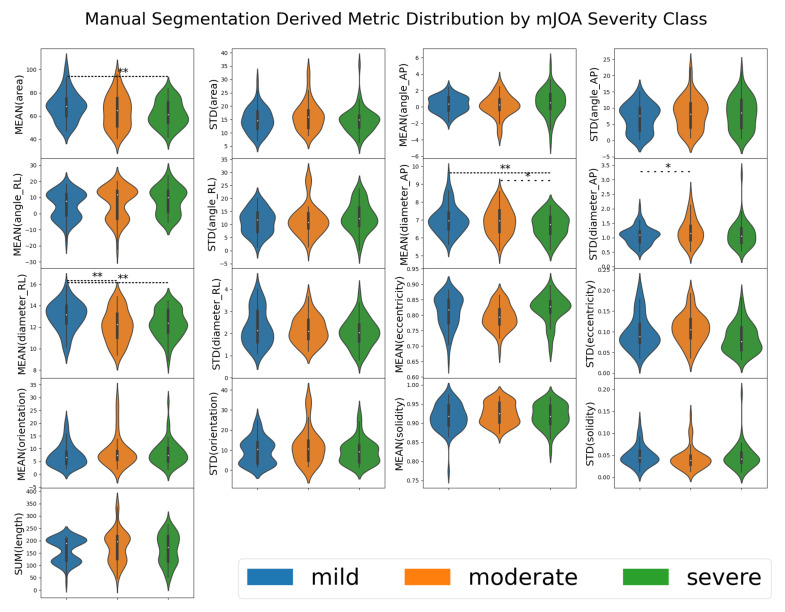
A violin plot of the distribution metrics extracted from manually segmented spinal cord images for 50 patients via the SCT. Each box represents one of the metrics evaluated by SCT, with the results grouped by mJOA severity classes. When the metric for one mJOA severity class was significantly different from another mJOA severity class (as determined by one-way ANOVA using SciPy’s f_oneway function returning a *p*-value less than 0.05), a line denoting such is present. A single * with a sparse dotted line denotes p<0.05, ** with a tightly dotted line denotes p<0.01. Metrics were taken from automated SCT analysis [13] of segmentations from 195 spinal cord MRI images.

**Figure 4 jcm-10-00892-f004:**
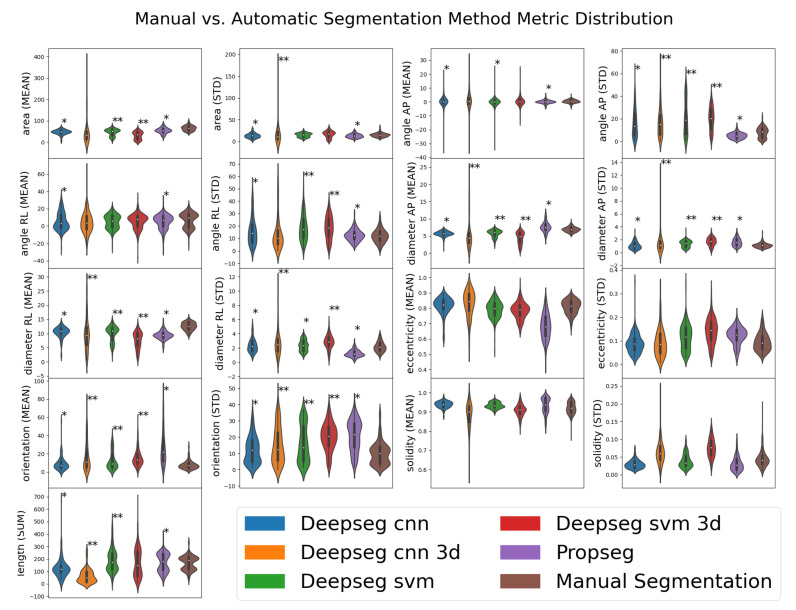
Visualized distributions of various metrics estimated by various segmentation methods for a subset of 50 patient records. Manual segmentation results are shown as the far-right distribution for each metric. Automated segmentation methods (not “Manual Segmentation”) are denoted with asterisks denoting how significantly different their distribution is from that of the “Manual Segmentation” distribution; ** for p<0.01, * for p<0.05, as evaluated by one-way ANOVA using SciPy’s f_oneway function (selected for its ease of implementation). Metrics taken from automated SCT analysis [13] of segmentations from 195 spinal cord MRI images.

**Figure 5 jcm-10-00892-f005:**
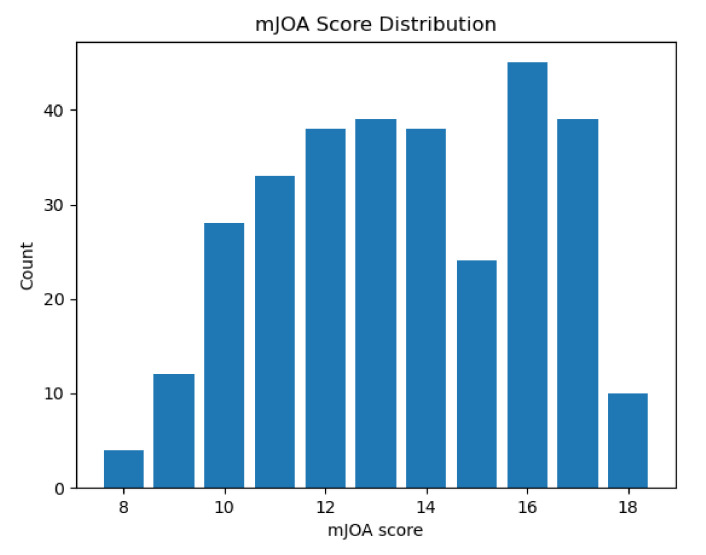
A box plot showing the number of individuals in our study with any given mJOA score. Although not quite ideal, this distribution is relatively balanced across the mid-range of mJOA scores. Note as well that extreme values (mJOA = 18 and mJOA = 8, 9) are rather rare, as would be expected given the acquisition method we used (data taken from those diagnosed with DCM who were undergoing initial assessment).

**Figure 6 jcm-10-00892-f006:**
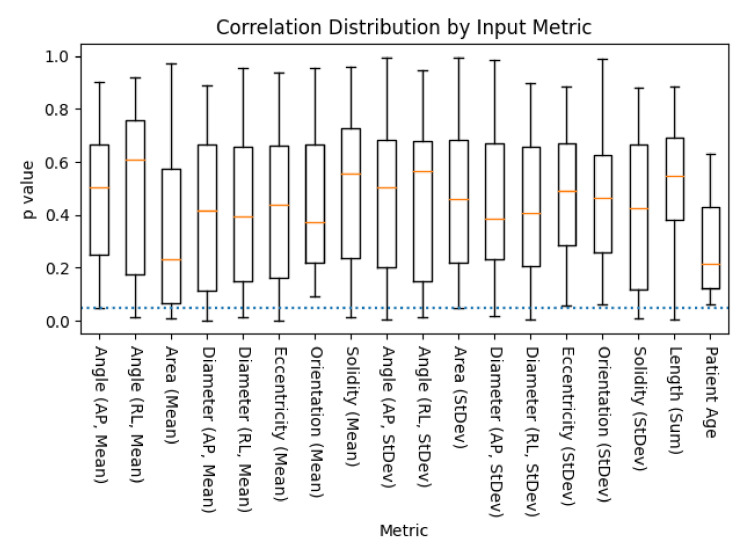
A box plot of the distribution *p*-values of metric to mJOA score correlations, across all combinations of acquisition contrast, orientation, and segmentation algorithm, as evaluated via SciKit-Learn’s ‘f_regression’ algorithm (lower is better). Age was included as a control, as it has been previously shown to be correlated with mJOA score [32]. The dotted blue line represents the threshold of significance for this study (p<0.05), with whiskers representing the maximum/minimum value of the set, or 1.5 times the inter-quartile range, whichever is shorter.

**Figure 7 jcm-10-00892-f007:**
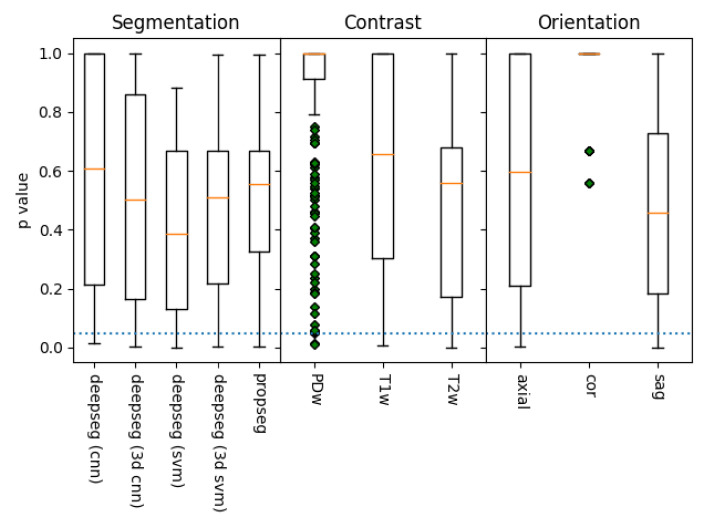
A box plot of the distribution *p*-values of metric to mJOA score correlations, grouped by acquisition contrast, orientation, and segmentation algorithm, as evaluated via SciKit-Learn’s ‘f_regression’ algorithm (lower is better). The dotted blue line represents the threshold of significance for this study (p≤0.05), with whiskers representing the maximum/minimum value of the set, or 1.5 times the inter-quartile range, whichever is shorter. Data points outside this range are denoted with green diamonds. Of the methods, it appears that segmentation using deepseg with a svm kernel provided the best results, as did those processed with a T2w contrast along the sagittal plane. However, all but coronal alignment appears capable of statistically significant metric extraction in at least some manner, though the PDw contrast is quite likely a fluke as well (due to its low sample size).

**Table 1 jcm-10-00892-t001:** Total number of segmentations resulting from each algorithm which were found to be “best-of-type” for a given patient. Ties were allowed, enabling one patient image to have up to two “best” segmentations.

Orientation	Contrast	Deepseg (cnn)	Deepseg (3d svm)	Deepseg (svm)	Propseg
sagittal	T2w	2	9	51	7
sagittal	T1w	0	0	6	29
sagittal	PDw	0	0	0	1
axial	T2w	13	0	63	0
axial	T1w	0	0	1	0
axial	PDw	0	0	0	0

**Table 2 jcm-10-00892-t002:** Variation of metric measures across mJOA severity classes in the manually segmented subset, summarized. Please note that the ’Mean/STD’ column denotes whether the metric used was the mean of the ’Metric’ column or the ’Standard Deviation’ of said ’Metric’ column. A visualized version of this data, alongside statistical assessments, can be found in Figure 3.

Metric	Mean/STD	Severe	Moderate	Mild
MEAN(area)	mean	62.223	64.066	68.393
MEAN(area)	std	10.999	14.206	12.574
STD(area)	mean	14.710	16.319	14.879
STD(area)	std	4.599	5.341	4.587
MEAN(angle_AP)	mean	0.585	0.193	0.320
MEAN(angle_AP)	std	1.595	1.338	0.964
STD(angle_AP)	mean	8.331	8.274	7.018
STD(angle_AP)	std	4.982	4.895	4.252
MEAN(angle_RL)	mean	8.029	6.554	5.188
MEAN(angle_RL)	std	8.354	9.779	8.244
STD(angle_RL)	mean	12.848	11.755	11.153
STD(angle_RL)	std	5.536	5.816	4.706
MEAN(diameter_AP)	mean	6.679	6.957	7.038
MEAN(diameter_AP)	std	0.666	0.778	0.787
STD(diameter_AP)	mean	1.109	1.231	1.073
STD(diameter_AP)	std	0.430	0.451	0.309
MEAN(diameter_RL)	mean	12.332	12.113	13.030
MEAN(diameter_RL)	std	1.308	1.520	1.339
STD(diameter_RL)	mean	2.049	2.175	2.300
STD(diameter_RL)	std	0.644	0.609	0.818
MEAN(eccentricity)	mean	0.820	0.795	0.811
MEAN(eccentricity)	std	0.045	0.040	0.051
STD(eccentricity)	mean	0.085	0.108	0.099
STD(eccentricity)	std	0.034	0.036	0.041
MEAN(orientation)	mean	8.222	8.692	7.331
MEAN(orientation)	std	4.680	5.956	4.338
STD(orientation)	mean	9.313	12.100	9.530
STD(orientation)	std	6.215	8.879	6.452
MEAN(solidity)	mean	0.920	0.925	0.917
MEAN(solidity)	std	0.031	0.028	0.034
STD(solidity)	mean	0.046	0.043	0.049
STD(solidity)	std	0.027	0.025	0.023
SUM(length)	mean	165.963	175.729	162.561
SUM(length)	std	59.023	63.725	46.554

**Table 3 jcm-10-00892-t003:** Variation of metric measures across automated segmentation methods. A visualized version of this data, alongside statistical assessments, can be found in Figure 4.

		Deepseg (cnn)		Deepseg (svm)			
Metric	Mean/Deviation	2d	3d	2d	3d	Manual	Propseg
MEAN(area)	mean	47.140	56.110	46.721	31.736	65.567	54.437
MEAN(area)	std	11.938	71.798	16.471	18.332	12.525	13.785
STD(area)	mean	13.528	24.376	14.993	16.562	15.366	13.336
STD(area)	std	5.627	37.167	5.033	7.937	4.841	4.717
MEAN(angle_AP)	mean	−0.099	−0.045	−0.173	0.273	0.374	0.039
MEAN(angle_AP)	std	4.842	8.273	3.535	3.917	1.283	1.381
STD(angle_AP)	mean	16.065	16.594	20.933	20.005	7.820	5.138
STD(angle_AP)	std	12.614	12.492	15.748	10.252	4.704	2.664
MEAN(angle_RL)	mean	5.600	4.448	5.036	5.475	6.639	5.166
MEAN(angle_RL)	std	10.255	12.174	7.908	8.534	8.556	8.035
STD(angle_RL)	mean	15.907	13.974	18.742	18.717	12.053	12.502
STD(angle_RL)	std	11.184	13.479	10.722	9.312	5.349	4.553
MEAN(diameter_AP)	mean	5.673	5.677	5.738	4.477	6.920	7.618
MEAN(diameter_AP)	std	0.835	4.638	1.102	1.752	0.736	1.498
STD(diameter_AP)	mean	1.107	1.863	1.362	1.690	1.127	1.617
STD(diameter_AP)	std	0.572	2.302	0.535	0.652	0.383	0.629
MEAN(diameter_RL)	mean	10.387	9.934	9.955	7.685	12.578	9.410
MEAN(diameter_RL)	std	2.019	6.107	2.701	2.921	1.423	1.537
STD(diameter_RL)	mean	2.346	2.828	2.353	2.834	2.189	1.243
STD(diameter_RL)	std	0.948	2.133	0.798	0.972	0.713	0.495
MEAN(eccentricity)	mean	0.815	0.829	0.792	0.784	0.810	0.683
MEAN(eccentricity)	std	0.057	0.086	0.055	0.054	0.046	0.084
STD(eccentricity)	mean	0.090	0.092	0.116	0.141	0.096	0.121
STD(eccentricity)	std	0.042	0.058	0.054	0.054	0.038	0.037
MEAN(orientation)	mean	9.424	17.098	12.619	15.474	7.805	27.025
MEAN(orientation)	std	8.231	16.024	9.849	9.238	4.850	18.971
STD(orientation)	mean	12.068	15.077	15.893	20.170	10.081	20.863
STD(orientation)	std	8.249	11.239	11.318	8.440	7.103	9.367
MEAN(solidity)	mean	0.938	0.883	0.934	0.908	0.920	0.933
MEAN(solidity)	std	0.017	0.070	0.016	0.031	0.031	0.041
STD(solidity)	mean	0.030	0.063	0.040	0.076	0.046	0.032
STD(solidity)	std	0.012	0.032	0.020	0.027	0.024	0.021
SUM(length)	mean	126.828	63.919	188.960	167.814	167.805	171.913
SUM(length)	std	80.717	57.264	92.204	112.005	55.064	72.697

**Table 4 jcm-10-00892-t004:** The attributes of our linear models fit on metric data, which was standardized to a common scale, but did not become harmonized by scanner used via CovBat. Orientation, contrast, and segmentation represent the acquisition methodology associated with the model. Features contains the list of features used to train the model, as selected by SciKit-Learn’s SelectFdr function.

Orientation	Contrast	Segmentation	Samples No.	Features	$r^{2}$
acq-axial	T2w	deepseg_cnn_3d	395	STD(angle_RL), MEAN(angle_AP)	−30.492
acq-sag	T2w	deepseg_svm	329	STD(angle_AP)	−29.873
acq-sag	T2w	propseg	308	MEAN(diameter_AP)	−30.576

**Table 5 jcm-10-00892-t005:** The attributes of our linear models fit on metric data which was standardized to a common scale and harmonized by scanner used via CovBat. Orientation, contrast, and segmentation represent the acquisition methodology associated with the model. Features contains the list of features used to train the model, as selected by SciKit-Learn’s SelectFdr function.

Orientation	Contrast	Segmentation	Samples No.	Features	$r^{2}$
acq-sag	T2w	deepseg_svm	329	STD(angle_AP), MEAN(angle_AP), STD(angle_RL)	−10.329
acq-sag	T2w	deepseg_svm_3d	329	MEAN(angle_AP), MEAN(diameter_RL)	−15.927
acq-sag	T2w	propseg	308	MEAN(orientation)	−25.549

**Table 6 jcm-10-00892-t006:** The attributes of logistic models fit on metric data, which was standardized to a common scale, but not and harmonized by scanner used via CovBat. Severity indicates the class attempting to be distinguished from all others (binary classification), while orientation, contrast, and segmentation represent the acquisition methodology associated with the model. Features contains the list of features used to train the model, as selected by SciKit-Learn’s SelectFdr function.

Severity	Orientation	Contrast	Segmentation	Sample No.	Features	AUC
severe	acq-axial	T2w	propseg	413	MEAN(eccentricity), STD(area)	0.713
severe	acq-sag	T2w	deepseg_cnn	269	STD(area)	0.519
moderate	acq-axial	T2w	deepseg_cnn	420	MEAN(area)	0.568
moderate	acq-axial	T2w	deepseg_svm_3d	420	STD(solidity)	0.549
mild	acq-sag	PDw	deepseg_svm_3d	27	MEAN(angle_RL)	0.920

**Table 7 jcm-10-00892-t007:** The attributes of logistic models fit on metric data which was standardized to a common scale and harmonized by scanner used via CovBat. Severity indicates the class attempting to be distinguished from all others (binary classification), while orientation, contrast, and segmentation represent the acquisition methodology associated with the model. Features contains the list of features used to train the model, as selected by SciKit-Learn’s SelectFdr function.

Severity	Orientation	Contrast	Segmentation	Samples No.	Features	AUC
severe	acq-sag	T2w	deepseg_svm_3d	329	MEAN(diameter_RL)	0.630
moderate	acq-axial	T2w	deepseg_svm_3d	420	STD(solidity)	0.538
mild	acq-sag	PDw	deepseg_svm_3d	27	STD(diameter_RL)	0.75
mild	acq-sag	T2w	deepseg_svm	329	STD(angle_RL)	0.558
mild	acq-sag	T2w	deepseg_svm_3d	329	STD(orientation), MEAN(eccentricity)	0.592

## Data Availability

The data are not publicly available due to its sensitive nature, to preserve the privacy of those enrolled in our study. However, anonymized data may be made available in the context of a data-sharing agreement in coordination with the corresponding author and appropriate research ethics approval.

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
