# Peer review of "Spinal Cord Morphology in Degenerative Cervical Myelopathy Patients; Assessing Key Morphological Characteristics Using Machine Vision Tools"

_jcm, 2021, doi:10.3390/jcm10040892_

Round 1

Reviewer 1 Report

The authors assessed key morphological characteristics using machine vision tools. Predicting the severity of DCM using image analysis is a very novel and interesting field. On the other hand, in this paper, the description is mainly in the research method on the analysis, and it is difficult for clinician to understand. In order for medical doctors to make use of this analysis in their daily clinical practice, they should be translated radiographical findings into language and summarized in the summary and in the beginning of discussions. The biggest problem of this study is that it was conducted in approximately 300 cases “diagnosed with DCM” in the specific region. Asymptomatic patients sometimes have severe stenosis, and the symptoms of myelopathy do not always correspond to imaging findings.1 This study looks the diagnosis and severity of DCM are subject to imaging, and the content of this research partly show the essence of DCM. Because spinal stenosis and myelopathy are distinct, it is necessary to carefully explain that the current study has seen a limited number of cases.

  1. Nakashima H, Yukawa Y, Suda K, Yamagata M, Ueta T, Kato F. Abnormal Findings on Magnetic Resonance Images of the Cervical Spines in 1,211 Asymptomatic Subjects. Spine. 2015;40(6):392-8.

Reviewer 2 Report

I congratulate the authors for this validation attempt. Indeed Machine Learning is showing promising tools in several fields where categorization and severity prediction are needed. By testing these tools one can assess their reliability and/or need to improve.

Though, they should aim more after concluding this retrospective analysis and endeavor to propose improvements to the existent tools.
